# The AAV capsid can influence the epigenetic marking of rAAV delivered episomal genomes in a species dependent manner

Adriana Gonzalez-Sandoval[1,2,5,7], Katja Pekrun [1,2,7], Shinnosuke Tsuji [1,2,6,7], Feijie Zhang[1,2], King L. Hung [3], Howard Y. Chang [3,4] & Mark A. Kay [1,2] ✉

Recombinant adeno-associated viral vectors (rAAVs) are among the most commonly used vehicles for in vivo based gene therapies. However, it is hard to predict which AAV capsid will provide the most robust expression in human subjects due to the observed discordance in vector-mediated transduction between species. In our study, we use a primate specific capsid, AAV-LK03, to demonstrate that the limitation of this capsid towards transduction of mouse cells is unrelated to cell entry and nuclear transport but rather due to depleted histone H3 chemical modifications related to active transcription, namely H3K4me3 and H3K27ac, on the vector DNA itself. A single-amino acid insertion into the AAV-LK03 capsid enables efficient transduction and the accumulation of active-related epigenetic marks on the vector chromatin in mouse without compromising transduction efficiency in human cells. Our study suggests that the capsid protein itself is involved in driving the epigenetic status of the vector genome, most likely during the process of uncoating. Programming viral chromatin states by capsid design may enable facile DNA transduction between vector and host species and ultimately lead to rational selection of AAV capsids for use in humans.

Adeno-associated virus (AAV) is a small single-stranded DNA (ssDNA) virus of the Parvoviridae family. While the discovery of a single capsid serotype and mechanisms related to basic AAV biology were described as early as 1960[1,2], in recent times recombinant AAV vectors have become the most popular vehicles for in vivo based gene therapy. Since the capsid sequence determines host tropism, substantial research over past decades has focused on creating vectors with novel properties to be effective in human gene therapy applications. Apart from isolating new AAV variants from natural reservoirs the capsid sequence has been modified either by rational design or directed evolution to develop capsids with specific properties[3–5]. One of the major limitations has been the discordance in transduction efficiencies

among species. Several studies have shown that rAAVs with capsids that can be used with high efficiency in preclinical mouse models are commonly found to be less efficient in non-human primate studies and/or human clinical trials[6]. In some cases, AAV capsids have been efficient in primates but not in rodents[7], thus making the use of a surrogate capsid for testing in preclinical mouse studies necessary. The unpredictability of serotype specificity adds time, cost, and uncertainty to the research process of developing effective gene-based therapeutics[8].

We decided to characterize the AAV transduction process in primate and rodent species, to better understand the primate selectivity observed with some AAV capsids. This study focuses on the AAV-LK03

[1]Department of Pediatrics, Stanford University, Stanford, CA, USA. [2]Department of Genetics, Stanford University, Stanford, CA, USA. [3]Department of Dermatology, Stanford University, Stanford, CA, USA. [4]Center for Personal Dynamic Regulomes and Howard Hughes Medical Institute, Stanford University, Stanford, CA, USA. [5]Present address: Encoded Therapeutics, South San Francisco, CA, USA. [6]Present address: Specialty Medicine Research Laboratories I, Daiichi Sankyo Co., Ltd., Tokyo, Japan. [7]These authors contributed equally: Adriana Gonzalez-Sandoval, Katja Pekrun, Shinnosuke Tsuji. ✉e-mail: markay@stanford.edu

capsid because of its current use in clinical trials[9]. AAV-LK03 originated from a capsid-shuffled library which had been selected in a xenograft humanized liver mouse model[7]. AAV-LK03-mediated gene transfer results in poor transgene expression in mice but performs robustly in primates including humans[9]. In the present study we compare different stages of the AAV-LK03 transduction process between both species. We show that the inefficiency of AAV-LK03 transduction in mouse cells is not due to major differences in cell entry or nuclear accumulation of vector genomes but rather correlates with a lack of histone modifications related to active transcription. Our study identifies epigenetic regulation as part of the species selectivity of AAV capsids. Our hypothesis is supported by the observation that addition of a single amino acid in the AAV-LK03 capsid, which restores transduction efficiency in murine cells, is associated with the accumulation of active-related epigenetic marks. Our results support previous findings that transduction efficiency is defined not only by the number of vector copies in the nucleus of the target cell, but also by transgene expression which can be capsid-dependent[10]. This study has important ramifications for AAV capsid development as it reveals the importance of developing AAV capsids that enable formation of active chromatin in the cargo vector genome.

## Results

### AAV-LK03 derived genomes internalize but do not express in rodent cells

To identify the mechanism of preferential activity of AAV-LK03 in primate cells, we assayed different steps of the transduction process in vitro and in vivo (Fig. 1). We used AAV-DJ (a chimeric capsid isolated from a capsid shuffled library, selected on human hepatoma cells) as a control capsid, as it is known to transduce both primate and rodent cells[11]. Huh7 and Hepa1-6 hepatoma cell lines were utilized as representative cells of human and mouse origin, respectively. A constitutive luciferase expression cassette (CAG-Luciferase) was packaged with AAV-LK03 and AAV-DJ capsids and the two cell lines were transduced with the resulting rAAV vectors.was packaged with AAV-LK03 and AAV-DJ capsids and the two cell lines were transduced with the resulting rAAV vectors.

Consistent with previous results[7], AAV-LK03 resulted in 100x lower luciferase activity in the mouse Hepa1-6 cells as compared to the human Huh7 cells, while AAV-DJ showed similar luciferase activity in both cell lines (Fig. 1a). Luciferase activity was shown to correlate with relative luciferase mRNA levels (Fig. 1b). In line with the expression result, we found that the vector copy number in whole cell lysates and fractionated nuclear lysates (Fig. 1c, d) was only marginally reduced in the mouse cells as compared to the human cells. AAV-DJ delivered vector DNA showed only slightly reduced expression in mouse vs humans with similar DNA vector concentrations in both cell lines. The 100x difference in luciferase activity between the human and mouse cells did not quite reach statistical significance due to high variation among the replicates in that experiment (Fig. 1a). However, we performed a more expansive study to confirm the 100x difference in AAV-LK03 mediated transgene expression (luciferase activity and luciferase mRNA) with similar levels of AAV-LK03 DNA uptake between mouse and human cells (Supplementary Fig. 1e, f, g). We confirmed that our observations were not unique to these particular cell lines or expression cassette, as similar results were obtained using vectors with different promoters and transgenes in various human and mouse cell lines (Supplementary Fig. 1a, b, c, h). Similar results were obtained in vivo. The livers of mice transduced with the AAV-LK03 or AAV-DJ CAG-Luciferase vector showed only a 3-fold difference in nuclear vector copy number (Fig. 1i). In contrast, the transgene mRNA and protein expression was >100x less in AAV-LK03 vs AAV-DJ treated animals (Fig. 1f, g).

As part of the transduction process, before the transgene can be expressed, the individually packaged plus or minus ssDNA AAV genome is released from the AAV capsid. The ssDNA is then converted into an episome, primarily as double-stranded DNA (dsDNA) circles[12,13]. We compared the uncoating efficiency of AAV-LK03 between species by treating transduced cells with DNAse I prior to DNA extraction to degrade the uncoated vector genomes (Fig. 1e). We found that a similar proportion of vector genomes was encapsidated and thus protected from degradation in the +DNAse I condition in both species. Southern Blot analysis of transduced mouse liver (Fig. 1j) revealed that the quantity and conformation of AAV-LK03 derived episomes were comparable to those derived from the AAV-DJ capsid, suggesting that vector uncoating and episome formation was not responsible for the poor expression from AAV-LK03 delivered genomes in mouse liver. This was further confirmed when we used a self-complementary (sc) rAAV vector to circumvent the need for double strand formation, and observed a similar level of discordance in transduction efficiency (transgene expression) between species (Supplementary Fig. 1d). Thus, episome formation was unlikely to be the main restriction factor for the observed species specificity of AAV-LK03.

### AAV-LK03 delivered genomes lack active histone marks in mouse cells

Given the comparable number of episomes derived from both AAV-LK03 and AAV-DJ in vivo, we reasoned that the ~3-fold decrease in nuclear vector genomes cannot be the primary reason for the >100-fold difference in mRNA or luciferase activity. We hypothesized that differences in chromatin structure and/or histone modification between the genomes delivered by different capsids might influence the silencing or activation of episomal vectors. We performed Cut&Tag[14] to quantify the genome-wide landscape of histone modifications, including those related to active transcription (H3K4me3 and H3K27ac) and those related to transcriptional repression (H3K9me3 and H3K27me3)[15]. Enrichment of H3K4me3 epigenetic modification on the gAAV (vector genome) delivered by AAV-LK03 or AAV-DJ to human or mouse cells is shown in Fig. 2a. AAV-LK03 delivered genomes were depleted for this active-related histone modification in the gene body region, with only a minor enrichment around the ITRs in mouse cells. As a control, enrichment of H3K4me3 within the albumin (Alb) gene was similar in all samples transduced with either capsid, regardless of the species transduced (Supplementary Fig. 2a). Looking at all histone modifications and conditions (Fig. 2b), plotted as coverage (which considers enrichment, read length and genomic size region), we found that AAV-LK03 delivered genomes were depleted for both histone modifications linked to active transcription (H3K4me3 and H3K27ac) in mouse cells, while the repressive histone modifications (H3K9me3 and H3K27me3) were less enriched on AAV delivered genomes in both species irrespective of the capsid used. This suggests that the reduced transcription from AAV-LK03 does not stem from the accumulation of repressive-related histone modifications, but rather from the lack of active-related histone modifications.

To interrogate if the low enrichment of histone modifications in mouse cells transduced with AAV-LK03 was caused by a lack of core histones, we performed Cut&Tag targeting core histones H2A, H3 and H4 (Supplementary Fig 2b). AAV-LK03-delivered genomes showed comparable levels of core histones in all experimental conditions, suggesting proper nucleosome assembly on AAV-LK03 delivered genomes in mouse cells.

While the Cut&Tag coverage plots of gAAV were normalized by total DNA coverage, there was a significant difference in gAAV reads obtained from human and mouse cells independent of the capsid. In order to make sure our data was not biased due to this parameter, we used a second normalization step. To do this, we created Tn5 DNA libraries for all conditions (Supplementary Fig. 2c), similar to what an input sample would be for chromatin immunoprecipitation. The mean coverage value of Tn5 libraries was used to divide the Cut&Tag coverage of all targets.

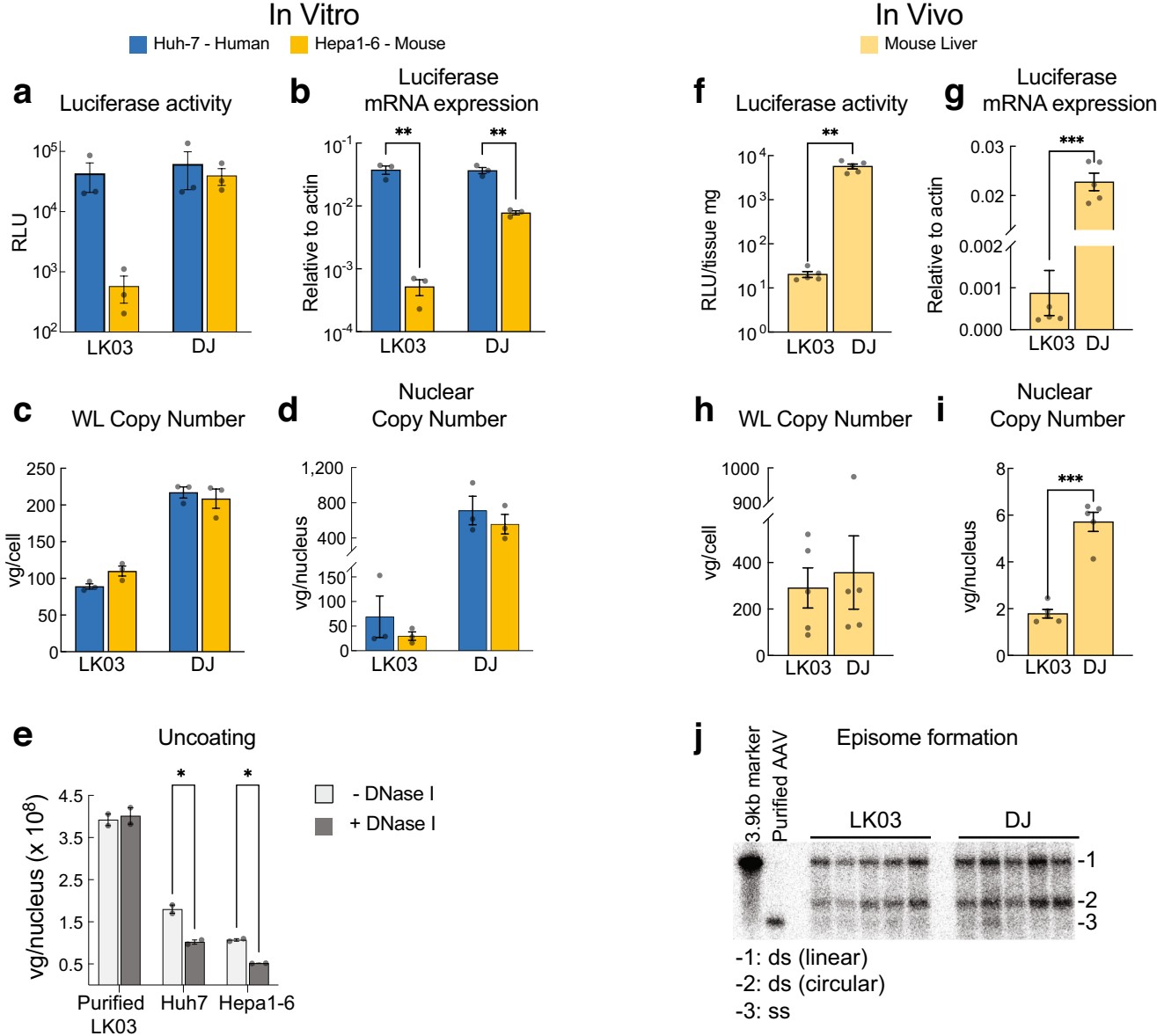

**Fig. 1 | Characterization of AAV-LK03 transduction in human and mouse hepatoma cells. a–e** In vitro assays using cell lines 48 h post transduction, (**f–j**) in vivo assays using liver tissue 3 days post rAAV delivery (i.v. tail vein). **a**, **f** Luciferase activity quantification. **b**, **g** Relative quantification of Luciferase mRNA by qRT-PCR. **c**, **h** Quantification of vector genomes by qPCR in whole cell lysate (WL) or (**d**, **i**) in nuclei. **e** Determination of the uncoating efficiency of AAV-LK03 by quantification of vector copies in DNase I treated vs untreated nuclear extracts. **j** Southern blot analysis in liver tissue probing for the luciferase gene. The

uncropped Southern blot image is provided in Source Data File 1. Statistics were performed using multiple unpaired Welch $t$ tests (**a–e**) or unpaired $t$ test with Welch's correction (**f–i**). Only statistically significant differences are indicated. Statistic $p$-value * <0.05, ** <0.01, *** <0.001. $N$ = 3 biologically independent samples, except (**e**) with 2. Data are presented as mean values +/− SEM. Raw data for the graphs are provided in Source Data File 1. Detailed statistics for each graph are provided in Source Data File 2.

## Single amino insertion into AAV-LK03 capsid enhances vector-mediated transgene expression and episomal active-related epigenetic marks in mice

Capsid sequence alignment revealed loss of a glycine or threonine residue from the highly variable region around serine 264 to serine 267 (S264–S267) in capsids with preferential activity in primates over mice[16] (Supplementary Fig. 3a). We created AAVLK03-265insT by inserting a threonine residue into AAV-LK03, similar to what has been described for the closely related AAV3B[14] and confirmed improved mouse cell tropism for this variant (Supplementary Fig. 3b, c). However, this particular insertion leads to a considerable increase in nuclear vector copies in mouse cells and, therefore, may obscure mechanisms of capsid-specific species tropism other than the difference in DNA copies. (Supplementary Fig. 3d), which would hamper our

efforts to elucidate the mechanisms responsible for the observed capsid-specific species tropism. Thus, we created AAV-AM by inserting a glycine residue at position 265 based on AAV-LK03 protein sequence. According to a predictive structure model of this new variant compared to the parental AAV-LK03 capsid (AAV3B) crystal structure (Supplementary Fig. 3e), most of the capsid structure remains unchanged, except for an extension of the VR-I loop which allows for closer proximity of the adjacent alanine and serine side chains to the histidine side chain which is six amino acids downstream. Delivery of the luciferase expression cassette using the AAV-AM capsid restored luciferase expression in mouse cells to levels similar to those observed in human cells. Expression as measured by transcript or luciferase activity levels were also comparable to those achieved with AAV-LK03 in human cells (Fig. 3a, b). The enhanced expression was not the result

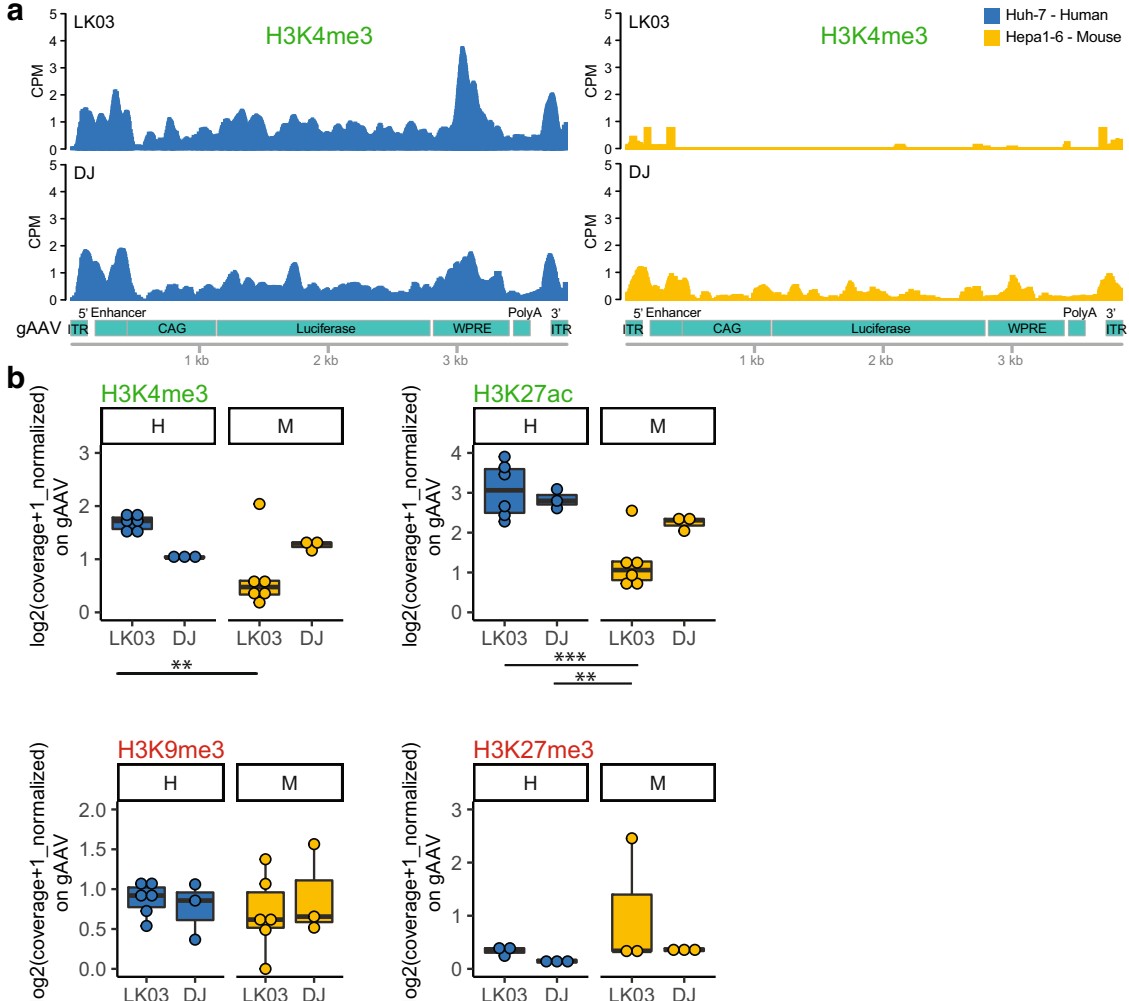

**Fig. 2 | Species-specific histone modifications observed in rAAV genomes delivered with the AAV-LK03 capsid into mouse vs human cells. a** H3K4me3 normalized (CPM) signal on AAV genome obtained from the Cut&Tag assay and next generation sequencing for Huh7 human and Hepa 1–6 mouse cell lines transduced with AAV-LK03 and AAV-DJ, as indicated. **b** Cut&Tag boxplots of normalized coverage on the AAV genome, for histone epigenetic modifications - H3K4me3 and H3K27ac (active transcription), H3K9me3 and H3K27me3 (inactive transcription) for Huh7 human and Hepa 1–6 mouse cell lines transduced with AAV-LK03 and AAV-DJ. Statistics were performed with 2 way ANOVA and only

statistically significant differences are indicated. Statistic *p*-value * <0.05, ** <0.01, *** <0.001. H = human, M = mouse. *N* = 3 biologically independent samples. Boxplots display the median (thick bar), two hinges (lower and upper hinges correspond to the first and third quartiles) and two whiskers. The upper whisker extends from the hinge to the largest value no further than 1.5 *IQR (inter-quantile range) from the hinge. The lower whisker extends from the hinge to the smallest value at most 1.5 *IQR of the hinge. Raw data for the graphs are provided in Source Data File 1. Detailed statistics for each graph are provided in Source Data File 2.

of increased vector copy numbers (Fig. 3c), which were nearly identical between AAV-AM and AAV-LK03 in mouse cells.

We also observed enrichment of active histone marks on vector genomes delivered by AAV-AM, both for mouse as well as for human cells (Fig. 3d). The respressive histone modifications H3K9me3 and H3K27me3 were comparable to the other serotypes (Supplementary Fig. 4a). Core histones were also present in vector genomes delivered by AAV-AM (Supplementary Fig 4b).

Importantly, unlike AAV-LK03 treated animals, AAV-AM treated mice had significantly higher levels of luciferase activity and transcript levels (Fig. 3e, f, g). In contrast, AAV-AM versus AAV-LK03 treated mice had only ~3-fold higher number of nuclear vector genomes (Fig. 3h). We also performed Cut&Tag on mouse liver for the three different capsids. Enrichment of active-related histone marks was observed for AAV-AM and AAV-DJ, but to a lesser degree for AAV-LK03 delivered genomes (Fig. 3i). As a reference, the *Albumin* gene enrichment of H3K4me3 epigenetic mark was similar in all mouse liver samples transduced with either capsid (Supplementary Fig. 5). Enrichment of inactive-related histone modifications (Supplementary Fig. 6a), and

core histones (Supplementary Fig. 6b) were comparable between AAV-AM and AAV-DJ, while AAV-LK03 delivered genomes exhibited lower levels of enrichment.

An approach to study transcriptional differences was carried out using Cut&Tag on transduced human and mouse cell nuclei using an antibody directed against Pol II. As shown in Supplementary Fig. 7, AAV-LK03 delievered vector compared to AAV-DJ and AAV-AM delivered to mouse cells showed considerably lower levels of Pol II occupancy across the whole vector genome as compared to human cells. This result is consistent with the observed lower levels of transcripts and protein expression for AAV-LK03 packaged vectors in mouse cells as compared to human cells.

Our data strongly support the hypothesis that the failure of AAV-LK03 to efficiently transduce rodent derived cells and tissues is related to a lack of transcription-permissive histone modifications associated with the episome that results in reduced RNA-polymerase II occupancy and reduced vector-mediated transcription. In contrast, the AAV-AM capsid, which contains a single amino acid insertion as compared to the AAV-LK03 capsid allowed the vector genome to accumulate

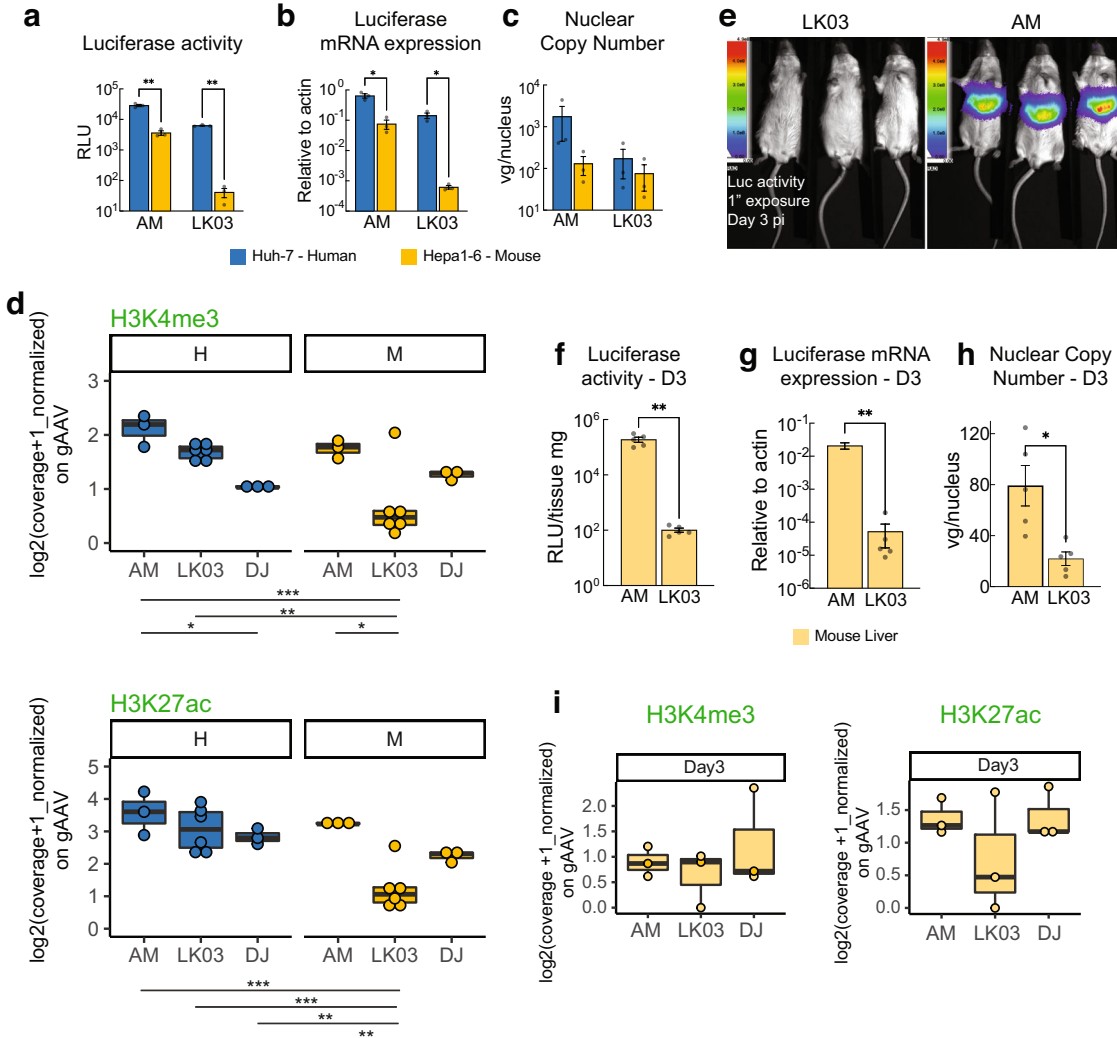

**Fig. 3 | A single glycine amino acid insertion in AAV-LK03 (AAV-AM) enables mouse efficient transduction, including active-related epigenetic profile in vitro and in vivo. a** In vitro 48 h transduction luciferase activity assay. **b** Luciferase transcript quantification. **c** Nuclear copy number quantification in Huh7 human and Hepa 1-6 mouse cells transduced with AAV-LK03 and AAV-AM by qPCR. **d** Cut&Tag boxplots of normalized coverage on AAV genome of active-related histone marks in vitro. **e** In vivo luciferase activity assay 3 days post intravenous injection of AAV-LK03 and AAV-AM. **f** Liver tissue luciferase assay. **g** Luciferase transcript quantification. **h** Nuclear copy number qPCR quantification 3 days post intravenous vector injection. **i** Cut&Tag boxplots of normalized coverage on AAV genome of active-related histone marks in vivo. Statistics were performed using multiple unpaired Welch *t* tests (**a–c**) or with unpaired *t* test with Welch's correction (**f–h**). For (**d**) a 2 way ANOVA, for (**i**) an ordinary one way ANOVA. Only statistically significant differences are indicated. Statistic *p*-value * <0.05, ** <0.01, *** <0.001. H = human, M = mouse. *N* > = 3 biologically independent samples. Data are presented as mean values + /− SEM in bargraphs. Boxplots display the median (thick bar), two hinges (lower and upper hinges correspond to the first and third quartiles) and two whiskers. The upper whisker extends from the hinge to the largest value no further than 1.5 *IQR (inter-quantile range) from the hinge. The lower whisker extends from the hinge to the smallest value at most 1.5 * IQR of the hinge. Raw data for the graphs are provided in Source Data File 1. Detailed statistics for each graph are provided in Source Data File 2.

activating epigenetic marks allowing more robust transcription and transgene expression in mouse cells as well as in human cells. Taken together our data support a model (Fig. 4a) whereby an AAV-capsid based on its sequence/structure differentially interacts with host proteins during the uncoating process to drive the epigenetic state of the vector episome.

### Dynamic change in the population of histone containing vector episomes in mouse liver

We sought to examine the stability of AAV episomes and their epigenetic state over time. Since episomal DNA is lost during cell division in cell culture studies, we examined the enrichment of histone modifications and core histones in quiescent mouse liver harvested 15 days post injection with CAG-FLuc expression vectors packaged with the three different capsids. Comparing day 15 vs day 3 harvested livers,

luciferase expression increased 760-, 650-, and 14- fold for AAV-LK03, AAV-DJ, and AAV-AM respectively (Fig. 4b, day 15 luciferase activity in Supplementary Fig 8a, day 3 luciferase activity in Fig. 1f and Fig. 3f), yet the gAAV nuclear copy number for AAV-AM, and AAV-LK03 was reduced 10-, and 100- fold, respectively during the 12-day interval while it increased slightly for AAV-DJ (Fig. 4c, Day 15 nuclear copy in Supplementary Fig 8b, day 3 nuclear copy in Fig. 1l and Fig. 3h).

However, we observed an increase in coverage of histone modifications on AAV-LK03 delivered vector genomes while modifications on genomes that had been delivered with the other two capsids either increased only marginally or even decreased over time (Fig. 4d). For core histones, we observed a consistent decline in coverage on genomes delivered with AAV-AM and AAV-DJ capsid while AAV-LK03 capsid delivered genomes showed an increase in coverage (Supplementary Fig. 8c). Interestingly, AAV-LK03 delivered genomes at day 15

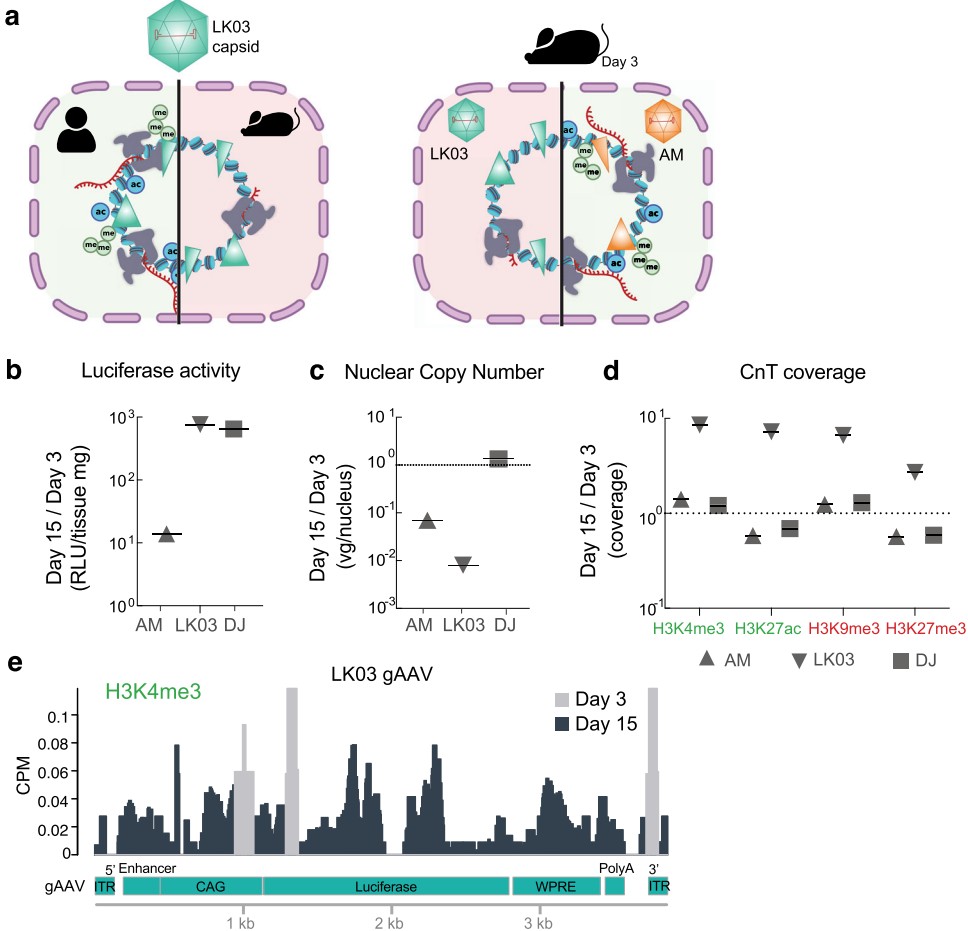

**Fig. 4 | Species selectivity of AAV-LK03 mediated transgene expression involves the formation of active-related histone epigenetic marks. a** Schematic representation of results (created with BioRender.com). AAV-LK03 derived genomes were enriched for active-related epigenetic marks in human, but not mouse cells. After adding a single amino acid to the AAV-LK03 capsid (AAV-AM), transduction was efficient in mouse cells and liver. Active-related epigenetic marks were enriched in AAV-AM transduced mice. **b–d** Ratio comparisons of in vivo assays on mouse liver tissue at Day 15 vs Day 3 post intravenous vector injection: (**b**) Luciferase activity, (**c**) Nuclear copy number RT-qPCR, (**d**) Tn5 normalized Cut&Tag coverage of histone epigenetic marks as indicated. **e** Comparison of Cut&Tag H3K4me3 normalized (CPM) signal on the AAV vector genome at different days post injection as indicated. Raw data for the graphs are provided in Source Data File 1.

post injection contained histone modifications across the entire gAAV, in contrast to the pattern observed at day 3 (Fig. 4e and Supplementary Fig. 8d). We speculate that perhaps the AAV-LK03 delivered genomes are lost because they do not form stable structures in mouse cells and are therefore degraded over time, leaving only a few genomes that had been chromatinized protecting them from degradation (Supplementary Fig. 8b, e). Therefore, the AAV-LK03 derived luciferase expression in mouse liver is likely derived from the low number of vector copies that remain and and are highly chromatinized and epigenetically modified.

## Discussion

The discordance observed by us and others regarding AAV-transduction efficiency between species has contributed to some of the difficulties in gene-based therapeutics. The use of AAV-LK03 required the use of surrogate capsids in preclinical rodent testing, which is not optimal. Our results further exemplify the importance of defining transduction and should include the measurement of transgene expression either by mRNA and/or protein production, as measuring vector genomes alone can be misleading. We had previously shown that naked DNA plasmids delivered into mouse liver cells via hydrodynamic transfection form nucleosomes and become associated with histones[17] so it is not surprising that something similar occurs with

rAAV particles once they uncoat in the nucleus and are converted to double-stranded episomes[18,19].

The AAV-AM capsid is a derivative of the primate selective AAV-LK03 capsid harboring a single amino acid insertion, which delivers a similar number of nuclear vector genomes and ultimately similar amounts of dsDNA episomes in murine cells in vitro or in vivo while providing >100x increased transgene expression. Using Cut&Tag to uncover epigenetic marks in a high-throughput format, we were able to establish that the differential expression patterns were related to the relative enhancement in transcriptionally permissive histone marks.

These results show that the capsid proteins play a role in establishing the chromatin state of AAV genomes. Early studies described AAV-2 capsids with single amino acid substitutions can internalize, uncoat and form ds-DNA episomes but do not express the transgene[10]. Two of the three mutant AAV-2 capsids showed enhanced transgene expression when a scAAV vector was used to package the vector and it was suggested these mutants were deficient in ss to ds-DNA conversion. Another mutant was also able to form ds episomal vector genomes and still did not express. In our study, scAAV-LK03, DJ and AM showed similar small enhancements in transgene expression in mouse cells suggesting the block in AAV-LK03 transduction was not related to a block in conversion to dsDNA. Taken together with our DNA analyses, the block in AAV-LK03 mediated transgene expression occurs

after dsDNA formation. It is not yet known if the results from the AAV-2 point mutant studies are related to changes in epigenetic chromatin formation and hence mechanistically related to our findings.

We speculate that the type of epigenetic marks placed on the vector genomes is dependent on how the sequence and/or structure of the capsid proteins differentially interact with chromatin modifiers, depending on species and/or various cell types. Recent evidence that AAV-mediated transgene expression comes from the Das group[20] who showed general epigenetic regulation in the silencing of AAV derived genomes by NP220, a host ds-DNA binding protein and the HUSH complex[20]. Interestingly, when the HUSH complex was knocked out in cells, AAV-mediated transgene expression was increased across a variety of different AAV serotypes tested, and this was associated with a decrease in transcriptionally silencing histone marks whereas in our studies enhanced expression was associated with an enrichment of transcriptionally active histone marks. Furthermore, Schreiber and colleagues[21] found that two components of the U2 snRNP spliceosome, PH5A and SF3B1 inhibit AAV-vector mediated transcription independent of their role in splicing regardless of the AAV serotype used. Both of these proteins have been implicated in chromatin remodeling, possibly by affecting histone modifications and deposition[22,23].

Unusual expression patterns after AAV transduction have been observed in other studies as well. For example, various capsids have differential effects on expression from various promoters used in the central nervous system[24]. In a humanized liver model AAV-DJ transduction patterns were different in the human xenograft vs mouse resident hepatocytes. In contrast to the mouse hepatocytes, a significant proportion of the human hepatocytes harbored AAV-positive nuclei despite a lack of transgene mRNA expression[25]. Recently, human liver biopsies from patients treated with a AAV-5 human FVIII showed variation between and within individuals for vector DNA copy number, DNA positive hepatocytes and mRNA expression[26]. In fact, one non-responder had intact vector and vector positive hepatocytes but little hVIII mRNA. Interestingly, this patient had no hepatic *HDAC9* and reduced *PH5A* expression. It is possible that higher mammals have evolved more sophisticated processes for episomal epigenetic regulation and perhaps there are polymorphic variant genetic factors within individual primates that may in part explain the wide range of responses in AAV-mediated gene transfer between individuals. Interestingly, differences in ATAC seq profiles and transgene expression have recently been observed in mice treated with an oversized AAV5 vector produced in insect versus mammalian cells[27]. Taken together, the studies suggest potential differences in capsid-mediated epigenetic regulation of rAAV delivered gene transfer vectors. Further investigation will be needed to understand the detailed mechanism by which the capsid proteins influence the deposition of various modified histones and perhaps other chromatin modifiers on the vector genome.

Our results underscore the importance of histone association and epigenetic regulation of vector genomes, resulting in retention in the nucleus and active transcription due to accumulation of activating histone marks. We propose that the nuclear uncoating process and histone association with the vector genome is a crucial step in the transduction mechanism and is dependent on the capsid sequence. Further understanding on how the capsid interacts with specific host nuclear factors will perhaps allow for improvements in designing future gene transfer vectors as well as being able to make better predictions when moving from preclinical animal to human clinical trials.

## Methods

### Ethical statement

All animal work was performed in accordance with the guidelines for animal care at Stanford University and approved by the APLAC committee protocol 13545.

### Vector plasmids

The AAV vectors containing ITR sequences used in this study are based on AAV type 2 backbone. Most experiments use rAAV vectors expressing Firefly Luciferase (FLuc) under control of a CAG promoter (pAAV-CAG-FLuc). The cloning of pAAV-CAG-FLuc (Addgene Cat# 83281) plasmid has been described previously[28].

An experiment uses rAAV vectors expressing FLuc under EF1a (core) promoter (pAAV-EF1a-FLuc). The pAAV-EF1a-FLuc plasmid was generated by replacing the TdRed sequence in pAAV-EF1a-TdRed (see below) with the FLuc gene with the regulatory WPRE region from pAAV-CAG-FLuc (Addgene Cat# 83281). Restriction enzymes BamHI-HF and SalI-HF were used to release the 3.4 kb vector band from pAAV-EF1a-TdRed as well as the 2.3 kb FLuc-WPRE sequence from pAAV-CAG-FLuc. Both fragments were ligated with Hi-T4 ligase and transformed into One Shot Stbl3 competent E.coli. The pAAV-EF1a-TdRed vector was generated by replacing the RSV promoter region in plasmid pAAV-RHB[29] with the first 212 nucleotides of the EF1a promoter from plasmid pAAV-EF1a-FLuc-WPRE-HGHpA (Addgene Cat# 87951) and the hAAT transgene with the TdRed sequence obtained from plasmid pAAV-CAG-tdTomato (Addgene Cat# 59462).

An experiment uses rAAV vectors expressing FLuc under CMV promoter (pAAV-CMV-FLuc). The pAAV-CMV-FLuc plasmid was prepared by swapping the CAG promoter from pAAV-CAG-FLuc (Addgene Cat# 83281) with the CMV promoter from pCMV6-Entry (Origene Cat# PS100001). BamHI-HF and NdeI restriction enzymes were used for both plasmids, the 5.6 kb and 0.4 kb fragments were isolated, ligated with Hi-T4 ligase and transformed in One Shot Stbl3 competent *E.coli*.

An experiment uses rAAV vectors expressing GFP (pAAV-CAG-GFP (Addgene Cat# 37825)) or TdTomato (pAAV-CAG-TdTomato (Addgene Cat# 59462)) under CAG promoter.

An experiment uses self-complementary rAAV vectors expressing RLuc under CAG promoter from pscAAV-CAG-RLuc (Addgene Cat# 83280).

The packaging vector AAV-LK03 (AAV2 rep LK03 cap) has been described previously[4]. AAV-LK03insT and AAV-AM were created by inserting a Threonine or Glycine respectively immediately downstream of the Serine at position 264 of the AAV-LK03 capsid using the QuikChange in vitro mutagenesis kit (Agilent).

### AAV production

rAAV vectors were produced using a triple transfection protocol with PEI 25 K. Briefly, the night before transfection, 293 T cells were seeded onto 15 cm² dishes (7.5e6 cells each dish in 25 mL media). The following morning, when the cells had reached 80–90% confluency, a transfection mix containing 3.75 ug ITR vector, 3.75 ug rep-cap vector, and 12 ug pAd5 helper plasmid with 105 uL PEI 25 K (1 mg/mL in $H_2O$ ph 4.5, linear polyethylenimine, MW 25,000, Polysciences Inc) in 2 mL Opti-MEM (Thermo Scientific) was prepared. After mixing by pipetting and a 15–20 min incubation at RT the transfection mix was added to the cells, after which they were returned to the incubator. Transfected cells were harvested after 72 hrs by adding 0.5 mM EDTA (315 uL per dish) and dislodging the cells using a serological pipette. For large-scale rAAV production 60 dishes were transfected while 15 dishes were used for small-scale production runs. Purification of rAAV was performed using two consecutive rounds of CsCl gradient centrifugation for large-scale productions as described previously[28] or using an AAVpro Purification kit (all serotypes; Takara) for small-scale production, following the manufacturer's instruction. Purified rAAVs were stored in aliquots at −80 °C until use. rAAV genomes were extracted and purified using a QIAamp MinElute Virus Spin kit (Qiagen) and titered by qPCR with serial dilutions of a plasmid standard. qPCR was performed with 2uL of corresponding material, in duplicate using Apex qPCR Green Master Mix (Genesee Scientific) and a CFX384 Touch Real-Time PCR Detection System (Bio-Rad) using the following cycling conditions: 95 °C for 15 min, 45 cycles of 95 °C for 10 s, 60 °C for 10 s

and 72 °C for 10 s and one cycle of 95 °C for 10 s and 65 °C for 1 min and 65–97 °C (5 °C s⁻¹). The sequence information for primers used in qPCR is shown in Supplementary Data 1.

### Cell culture and transduction

Huh7 (JCRB0403) cells were purchased from JCRB, 293 T (CRL-3216) and Hepa1-6 (CRL-1830), SNU-499 (CRL-2234), PLC (CRL-8024), C3A (CRL-10741), Hepa-1c1c7 (CRL-2026), BNL (CRL-3308), AML12 (CRL-2254) cells were purchased from ATCC. Huh7, Hepa 1-6, Hepa-1c1c7, BNL lines were cultured in DMEM with 10% fetal bovine serum (FBS), 2 mM L-glutamine and 2 mM sodium pyruvate; additional 22 mM of HEPES buffer is added to 293 T cells. SNU-499 were cultured in RPMI-1640 medium. PLC, C3A were culture in EMEM with 10% FBS. AML12 were cultured in DMEM-F-12 with 10% FBS. Cells were seeded and transduced after 24 hrs with a MOI (multiplicity of infection) of 1e4 vg/cell, and incubated for another 48 hrs for most of experiments, unless indicated for 24 hrs.

### Animals and transduction

All animal work was performed in accordance with the guidelines for animal care at Stanford University and approved by the APLAC committee protocol 13545. BALB/c scid mice (Strain #:001803) were purchased from Jackson Laboratory. We used 6 week-old juvenile male mice. rAAV delivery was accomplished by tail vain injection with 3e11vg per mouse. Mice were housed at 18.3–23.9 °C, with 40–60% humidity on a 12-h light/12-h dark cycle. Food and water were given ad libitum.

At the end of each experiment, mice were anesthetized with isoflurane and perfused transcardially with PBS, and liver tissues were quickly collected and cut into several pieces. The tissues for mRNA extractions were immediately submerged in RNAlater solution (Sigma-Aldrich) and stored at 4 °C until use. For luficerase, gDNA or Cut&Tag assays, tissues were snap-frozen in liquid nitrogen and stored at −80 °C until use.

### Luciferase assays

Firefly luciferase assays in vitro were performed using the ONE-Glo Luciferase Assay System (Promega) following the manufacturer's instruction. Briefly, at indicated time points after rAAV transduction, 50 uL of the reconstituted substrate was added to the cells grown in 96-well plates and incubated for 10 min with gentle shaking. Luminescent activity was measured using a plate reader and Tecan i-control Microplate Reader Software and Tecan i-control™ Microplate Reader Software (v3.4.2).

For experiments shown in Supplementary Fig. 1e and 1h the firefly luciferase assays were set up in a slightly different format. On the night prior to transduction 48 well plates were seeded with 40,000 cells per well in a volume of 0.5 mL. On the next morning rAAV was added at an MOI of 10,000 vg/cell. 48 hrs post transduction the medium was aspirated, the cells were rinsed with DPBS, and 100 ul 1x Passive Lysis Buffer (Promega) were added to each wells. Cells were lysed at room temperature for 5 min, transferred into strip tubes, and cell debris was pelleted by a brief spin. 20 uL of the clarified cell lysate were transferred into solid white 96 well plates and 100 uL of reconstituted luciferase substrate (Luciferase 1000 Assay System, Promega) were added using a Veritas luminometer system with injector using the GloMax software. A setting of a 2-second measurement delay followed by a 10-second measurement read was chosen. A standard curve was set up by performing a series of 2-fold dilutions of recombinant Firefly luciferase (Quantilum, Promega) in Passive Lysis Buffer.

In vivo firefly luciferase imaging of mice was performed by intra peritoneal injection of 150 μg per g body weight D-Luciferin (Biosynth, catalog L-8220) and ventral luciferase readings using an Ami Imaging System.

Firefly luciferase assay of liver tissue was performed measuring tissue weight (ranging from 50–300 mg), homogenizing tissue in RINO

1.5-mL Screw-Cap tubes filled with stainless steel beads (Next Advance Inc, NC0542451) and 200 uL of 1x Passive Lysis buffer (Promega) using a bead homogenizer (Next Advance Bullet Blender Storm - BBY24M) at speed 8 for 3 min. Celll debris was pelleted by centrifugation at 9600 × g for 10 min at 4 °C and supernatants were collected as liver extract. 5 uL of liver extract were added to 100 uL of ONE-Glo Luciferase reconstituted substrate and incubated for 10 min with gentle shaking.

Luminescent activity was measured using a plate reader and Tecan i-control™ Microplate Reader Software (v3.4.2). Values were normalized by tissue weight. Three technical replicas were performed for each liver tissue. The *Renilla* luciferase assay was performed according to the kit instructions (Promega E2810). Further analysis was performed in Microsoft Excel (v2111).

### RNA extraction, cDNA preparation and RT-qPCR

Total RNA was extracted using a RNeasy micro plus kit (Qiagen) according to the manufacturer's protocol with on-column DNase treatment for 30 min.

For in vitro experiments, cultured cells transduced with rAAVs were collected by trypsinization and washed with PBS. At least 1e6 cells were used for RNA extraction.

Liver tissue samples stabilized in RNAlater solution (~100 mg) were homogenized in RINO 1.5-mL Screw-Cap tubes filled with stainless steel beads (Next Advance Inc, NC0542451) and 800 μL of RLT buffer (including β-mercaptoethanol) using a bead homogenizer (Next Advance Bullet Blender Storm - BBY24M). 600 uL of lysate were used for total RNA extraction.

cDNA was synthesized from 50–100 ng of total RNA using a High-Capacity RNA-to-cDNA kit (Life Technologies) according to the manufacturer's instructions.

qPCR was performed with 2 uL of 1:5 diluted cDNA, in duplicate using Apex qPCR Green Master Mix (Genesee Scientific) and a CFX384 Touch Real-Time PCR Detection System (BioRad CFX Maestro (vl.1)) using the following cycling conditions: 95 °C for 15 min, 45 cycles of 95 °C for 10 s, 60 °C for 10 s and 72 °C for 10 s and one cycle of 95 °C for 10 s and 65 °C for 1 min and 65–97 °C (5 °C s–1). Targets of interest were normalized against ActinB (human or mouse) as Delta-Delta Ct calculations. Sequence information for all qPCR primers is listed in Supplementary Data 1. Further analysis was performed in Microsoft Excel (v2111).

### gDNA extraction

Total gDNA was extracted using a QIAamp DNA Mini kit (Qiagen) in cultured cells and DNeasy Blood & Tissue kit (Qiagen) in liver tissue, according to the manufacturer's protocols with addition of RNase A treatment.

Cultured cells transduced with rAAVs were collected by trypsinization and washed with PBS. At least 1e6 cells were used for total gDNA extraction.

Snap-frozen liver tissue (~100 mg) was homogenized in RINO 1.5-mL Screw-Cap tubes filled with stainless steel beads (Next Advance Inc, NC0542451) and 200 uL of 1x Passive Lysis buffer (Promega) using a bead homogenizer (Next Advance Bullet Blender Storm - BBY24M) at speed 8 for 3 min. After centrifuging at 9600 × g for 10 min at 4 °C, supernatant was collected as liver extract. 100 uL of liver extract were used for total gDNA extraction.

### Nuclear gDNA extraction

Cultured cells transduced with rAAVs were collected by trypsinization and washed with PBS. Nuclei were isolated with NE-PER™ Nuclear and Cytoplasmic Extraction Reagents (ThermoFisher Scientific). Briefly, the cell pellet was resuspended in CER I buffer + Halt Protease Inhibitor cocktail (ThermoFisher Scientific) in a volume according to pellet size as per manufacturer's instructions. The suspension was vortexed,

incubated on ice for 10 min and vortexed again. Then, CER II was added according to the manufacturer's instructions, the lysate was vortexed, incubated for 1 min on ice and vortexed again. Nuclei were pelleted by 3 min centrifugation at 12,600 × g at 4 °C, washed with 100 uL of CER I buffer, and 3 times with 500 uL of cold PBS spinning for 1 min at 12,600 × g. After adding 200 uL of AL Buffer + 20 uL Proteinase K, the nuclear pellet was sonicated for 5 min and genomic DNA was extracted using the QIAamp DNA Mini kit (Qiagen) according to the manufacturer's instructions. The additional RNase A treatment was included in the protocol.

Snap-frozen liver tissue (-100 mg) was homogenized in RINO 1.5-mL Screw-Cap tubes filled with stainless steel beads (Next Advance Inc, NC0542451) and 1 ml of CER I buffer using a bead homogenizer (Next Advance Bullet Blender Storm - BBY24M) at speed 8 for 3 min. The extract was transferred into a fresh tube, incubated on ice for 10 min, and vortexed. Subsequently, 55 uL of CER II were added, the mixture was vortexed, incubated for 1 min on ice and vortexed again. Nuclei were pelleted by 3 min spin at 16,200 × g at 4 °C, the nuclear pellet was washed with 100 uL of CER I buffer, and 3 times with 500 uL of cold PBS, and spun for 1 min at 16,200 × g. Finally, 200 uL of AL Buffer + 20 uL Proteinase K were added, the nuclear pellet was sonicated for 5 min and nuclear gDNA was extracted following the protocol of the DNeasy Blood & Tissue kit (Qiagen) with addition of RNase A treatment.

### Vector copy number qPCR

Nuclear or whole lysate DNA was used from cultured cells or liver tissues. qPCR was performed with 2 uL of eluted DNA, in duplicate using Apex qPCR Green Master Mix (Genesee Scientific) and a CFX384 Touch Real-Time PCR Detection System (BioRad CFX Maestro (vl.1)) using the following cycling conditions: 95 °C for 15 min, 45 cycles of 95 °C for 10 s, 60 °C for 10 s and 72 °C for 10 s and one cycle of 95 °C for 10 s and 65 °C for 1 min and 65–97 °C (5 °C s⁻¹). Standard curves for each primer set were generated using serially diluted plasmid (for gAAV) or Human and Mouse Genomic DNA (for host) (Promega G1521 and G3091, respectively) and used for quantification. CFX Maestro Software was used for data analysis. Sequence information for all qPCR primers is listed in Supplementary Data 1. Further analysis was performed in Microsoft Excel (v2111).

### Uncoating - DNase I treatment

Cultured cells transduced with rAAVs were collected by trypsinization and washed with PBS. Nuclei were extracted as described in the previous section (Nuclear gDNA extraction), then the nuclear pellets were resuspended in 200 uL of NER buffer, vortexed for 15 s and placed on ice with continued vortexing for 15 s every 10 min for total 1 h. Subsequently, the lysates were centrifuged at 17,000 × g for 10 min, and the supernatants (nuclear extract) were transferred into new tubes and an equal amount of ice-cold PBS and DNase I buffer (10x) was added. The extracts were split into two tubes – 20 U DNase I (Thermo Fisher, 18068015) was added into one tube only. Reactions were incubated overnight at 37 °C. DNA was extracted following QIAamp DNA Mini kit (Qiagen) and eluted in 50 uL water. qPCR was performed with 2 uL of eluted DNA and the same parameters as described above in Vector copy number qPCR section. 1e9 vector copies of purified AAV treated in the same manner as the nuclei were used as control.

### Southern blotting

Nuclear gDNA was extracted from liver tissue as indicated above. Then, gDNA was digested overnight with XhoI (NEB) that does not cut in the vector, but in the host gDNA. Digested DNA was run in a 1% TAE agarose gel at room temperature overnight. After electrophoresis, the gel was washed with denaturing buffer (3 M NaCl and 400 mM NaOH) twice for 5 min, and DNA was transferred to an Amersham Hybond-XL membrane (GE Healthcare) using transfer buffer (3 M NaCl and 8 mM

NaOH) overnight. Membranes were washed with 2× saline sodium citrate (SSC) buffer for 5 min and blocked with UltraPure Salmon Sperm DNA (Thermo Fisher) in QuikHyb Hybridization Solution (Agilent Technologies) for 1 h at 65 °C. Probes for FLuc (500 bp) were generated using gel-purified PCR amplicons containing GFP sequence and a BcaBEST Labeling kit (Takara) and [α-32P]-dCTP (PerkinElmer), and probe hybridization were performed overnight at 65 °C with rotation. The membrane was washed with 2× SSC buffer and with 2× SSC containing 0.1% SDS at 65 °C. Signals were visualized using a Personal Molecular Imager System (Bio-Rad ChemiDoc Imaging Systems (vS.2.1 build 11)) and analyzed with Quantity One 1-D software v4.6.8 (Bio-Rad).

### Flow cytometry

Cultured cells transduced with rAAVs were collected by trypsinization, washed with PBS and resuspended in cold PBS. Cells were kept on ice and protected from light until analyzed. Singlet cells were determined based on forward scatter/side scatter (FSC/SSC) plot, and GFP + or TdTomato+ fractions were gated based on non-transduced cells, as negative control. Data was collected for 10,000 gated cells. The percentage of GFP + or TdTomato+ expressing cells was evaluated using BD LSRII flow cytometer with BD FACSDiva software, and data were analyzed using the FlowJo software package. An example for the gating strategy is shown in Supplementary Fig. 9.

### Structure analysis

The software SWISS MODEL (https://swissmodel.expasy.org/) was used to build a partial capsid structure and the image was created with PyMOL GLSL Version 1.20.

### Alignment of capsid sequences

Aminoacid sequence of specified capsids were loaded into Clustal Omega web-based for alignment.

### Cut&Tag

Reagents and protocol are commercially available by Epicypher (https://www.epicypher.com/products/epigenetics-reagents-and-assays/cutana-cut-and-tag-assays). Details about reagents and antibodies used for Cut&Tag can be found in Supplementary Data 2.

Briefly, cultured cells transduced with rAAVs were collected by trypsinization and washed with PBS. Nuclei were extracted with NE buffer and mixed with activated Concanavalin A beads. After successive incubations with primary antibody (overnight) and secondary antibody (for 30 min) in antibody and Digitonin150 buffer respectively, the beads were washed with Digitonin150 buffer and resuspended in Digitonin300 buffer with 2.5 uL of pA(G)-Tn5 for 1 h and washed with the same buffer. Incubations were performed at room temperature in low-retention PCR strip tubes (Epicypher). Tagmentation was performed for 1 h at 37 °C in Tagmentation buffer that provides MgCl₂. Beads were washed with TAPS buffer and DNA material was released by adding SDS Release Buffer and incubating at 58 °C for 1 h. Quenching of SDS was performed by adding SDS Quench buffer and PCR was performed directly on this material. Universal P5 and indexed P7 primer solutions were used (see Supplementary Data 3 for sequences), and 21 cycles of PCR were performed. Clean-up was performed with AMPure beads, eluted in 15 uL 0.1x TE buffer. Qubit and Bioanalyzer were used to verify library qualities before pooling samples for sequencing. The barcoded libraries were mixed to achieve equimolar representation aiming for a 10 nM final concentration. Sequencing was performed with Illumina HiSeq 4000, 150 cycles total per lane, 2 × 75 paired-end reads, depth of >2 M reads per sample.

For in vivo Cut&Tag, snap-frozen liver tissue (-100 mg) was homogenized in RINO 1.5-ml Screw-Cap tubes filled with stainless steel beads and 1 mL of NE buffer using a bead homogenizer (Next Advance Bullet Blender Storm - BBY24M) at speed 8 for 4 min. The homogenate

was transferred into to a 50 mL tube with 12 mL of NE buffer and incubated on ice for 10 min. Nuclei were pelleted by a 5 min centrifugation at $600 \times g$, the supernatant was decanted, and the pellet was resuspended in 1 mL of NE buffer. Nuclei were quantitated after Trypan Blue staining using an automated cell counter (Countess, Thermo Fisher) and diluted accordingly. 1e5 nuclei per target protein were used for Cut&Tag. The subsequent steps were identical to the protocol described above using nuclei from cultured cells.

### Tn5 normalizer - Illumina DNA prep
Cultured cells transduced with rAAVs were collected by trypsinization and washed with PBS. Nuclei were extracted for at least 5e5 cells with NE buffer following Epicypher Cut&Tag protocol. Total gDNA was extracted using a QIAamp DNA Mini kit (Qiagen) including RNAse treatment. The protocol for Illumina DNA prep was followed as per manufacturer's instructions with 100 ng of DNA as input, 30 min tagmentation and 8 cycles of PCR. The same indexed primers as for Cut&Tag were used for PCR amplification. Equal volumes of each barcoded library (each around 10 nM assessed by Bioanalyzer) were pooled and library quality and quantity was assessed using the Qubit and Bioanalyzer. Sequencing was performed with Illumina Novaseq SP100, 150 cycles total per lane (2 lanes total), $2 \times 75$ paired-end reads, depth of >15 M reads per sample.

For in vivo Tn5 assay, snap-frozen liver tissue (~100 mg) was homogenized in RINO 1.5-mL Screw-Cap tubes filled with stainless steel beads and 1 mL of NE buffer using a bead homogenizer (Next Advance Bullet Blender Storm - BBY24M) at speed 8 for 4 min. The homogenate was transferred into to a 50 mL tube with 12 mL of NE buffer and incubated on ice for 10 min. Nuclei were pelleted by a 5 min centrifugation at $600 \times g$, the supernatant was decanted, and the pellet was resuspended in 1 mL of NE buffer. Nuclei were quantitated after Trypan Blue staining using an automated cell counter (Countess, Thermo Fisher) and diluted accordingly. 5e5 nuclei per sample were used; subsequent steps were identical to the protocol described above using nuclei from cultured cells.

### Data processing and analysis
For all figures with barplots, data was processed in GraphPad Prism 9 (Version 9.3.1) software applying the statistical methods indicated in the figure legends. Error bars represent standard error of the mean, and $n = 3$ or more biological replicas, except for Fig. 1e and Supplementary Fig. 1c. Fig. 4b–d and Supplementary Fig 8c represent the ratios of means of at least 3 biological replicas.

For Cut&Tag a total of 42 (cells in culture) and 21 (liver tissue) different conditions with at least three replicas were processed and analyzed. Stanford SCG cluster was used to process all sequencing data.

Reads were trimmed with Trimmomatic-0.39 program for classical Illumina adapters. Aligned using Bowtie2 version 2.2.5 (https://sourceforge.net/projects/bowtie-bio/files/bowtie2/2.2.5/) with options:–local–very-sensitive-local–nounal–no-mixed–no-discordant–phred33 -I 10 -X 700; mapping was performed using the hg38 or mm10 build of the human and mouse genome respectively, merged with viral genome sequence (gAAV). The viral genome sequence is the same as plasmid pAAV-CAG-FLuc from ITR to ITR. PCR duplicates were eliminated with "MarkDuplicates" command of Picard version 2.23 (https://broadinstitute.github.io/picard/index.html).

Tracks were made as bedgraph files of normalized counts, using deeptools 3.3 (https://deeptools.readthedocs.io/en/develop/content/list_of_tools.html) bamCoverage with options:–binSize 1,–normalizeUsing CPM; track plots were made with R package Gviz 4.1. Coverage was calculated as SUM[(normalized count*(end-start position+1))/gAAV size] for each dataset of Cut&Tag and Tn5 normalizer. Importantly, ITR regions were removed from the calculation, as their coverage is overrepresented in all samples, perhaps due to recognition of ITRs from ssDNA. For all Cut&Tag datasets, coverage

was normalized as: gAAV coverage C&T/ Tn5 coverage of corresponding condition. Coverage boxplots with or without normalization were generated with R package ggplot2 4.1.2. Boxplots display the median (thick bar), two hinges (lower and upper hinges correspond to the first and third quartiles) and two whiskers. The upper whisker extends from the hinge to the largest value no further than 1.5 *IQR (inter-quantile range) from the hinge. The lower whisker extends from the hinge to the smallest value at most 1.5 * IQR of the hinge. Data beyond the end of the whiskers are called "outlying" points and are plotted individually.

### Statistics and reproducibility
No statistical method was used to predetermine sample size. No data were excluded from the analyses. The animals used were randonmly selected for each grouping. The Investigators were not blinded to allocation during experiments and outcome assessment.

### Reporting summary
Further information on research design is available in the Nature Portfolio Reporting Summary linked to this article.

## Data availability
All data needed to evaluate the conclusions in the study are present in the main text or the supplementary materials. Source data for all graphs are provided with this paper in Source Data File 1 (raw data and uncropped Southern Blot image) as well as Source Data File 2 (detailed statistics). Request for reagents should be directed to M.A. Kay. Plasmid vectors are available to all academic groups. Commerical requests will be considered by Stanford's Office of Licensing and Technology. All raw and processed sequence data have been deposited in GEO (Gene Expression Omnibus) and are available under accession number "GSE226268". Source data are provided with this paper.

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

## Acknowledgements

We thank W. Greenleaf and G. Marinov for helpful discussions. A.G-S. is grateful to S. Herrera, A. Gatto for help and support in bioinformatic analysis. A.G-S is thankful to I. Deshpande for help in modelling of AAV-AM capsid. We wish to acknowledge F. Puzzo and J. Rubin for help with statistical analysis. We also appreciate the help from H. Jang with bioinformatics and figure generation. This project was also supported by a NIH Shared Instrumentation Grant (S10-OD010580) from the National Center for Research Resources (NCRR) with significant contribution from Stanford's Beckman Center as well as the Stanford Small Animal Imaging Facility. The authors wish to acknowledge the Stanford Genomics Facility for performing high-throughput sequencing, as well as CMP and SCG facilities. This work was funded by NIH grant AI116698 awarded to M.A.K. A.G-S, K.P., S.T., and F.Z. were also supported by this grant. A.G-S. was furthermore supported by the Walter V. and Idun Berry Postdoctoral Fellowship.

## Author contributions

A.G-S., S.T. and M.A.K. designed the research. S.T, carried out initial experiments. F.Z. carried out mouse injections and liver harvesting. K.P participated in the in vivo luciferase assays, provided several reagents for AAV production and helped with figure generation. K. L. H. developed the initial computational analysis, which was later adjusted by A.G-S. A.G-S., S.T., K.L.H., analyzed data. H.C. provided discussion. A.G-S, K.P., and M.A.K. wrote the manuscript.

## Competing interests

A patent application for AAV-AM was filed by Stanford University where A.G. and M.A.K. are inventors on a provisional patent application (# 63/333,459). The remaining authors declare no competing interests.
