## [Peer Review File · Nature Communications]

REVIEWER COMMENTS

Reviewer #1 (Remarks to the Author):

This manuscript by Gonzalez-Sandoval and colleagues is focused on evaluating the interplay between AAV capsids and their genomes, which in turn is hypothesized to impact transduction efficiency. Briefly, the authors provide evidence that a capsid (AAV-LK03) isolated from primate liver (humanized mouse model) tissue through directed evolution, shows decreased transduction in murine hepatocytes and imparts distinct chromatin marks on the vector genome. Depleted histone marks generally associated with transcription are observed in case of the original capsid only in mice, but a single point mutation is associated with active marks on the vector genome presumably restoring transduction in both species. Overall, the study presents novel observations/phenomena potentially linking the capsid and epigenetic modifications of the vector genome. The approach utilized is balanced, toggling skillfully between capsid modifications and extensive interrogation of molecular fate of vector genomes in human and mouse liver models. The manuscript captures the essence of these seemingly unrelated events in AAV biology and their novelty as well as briefly outlines implications for human gene therapy. However, significant gaps in the data, experimental design/controls and overall conclusions are noted as outlined below.

1. First, the authors confirm that LK03 shows lower transgene and mRNA transcript expression in human hepatocarcinoma cells in vitro over the mouse cell line. This appears to correlate with lower H3K4me and H3K27Ac signal in murine vs human cell lines, and no change is seen with H3K9me3 and H3K27me. However, the current normalization method seems to be focused on read depth using a Tn5 normalizer method. No controls using isotype matched IgG have been shown/evaluated. Normalization to the latter signal will be important in addressing the extent of non-specific reads throughout the manuscript. This is particularly important since ITR related reads have been removed from the overall calculations (due to overrepresentation).

2. Second, the authors note that a single G insertion in LK03 (AAV-AM) increases transduction concomitant with H3K4me and H3K27Ac reads increasing both in vitro and in vivo. However, no mRNA levels are shown in vitro or in vivo (Figure 3). Since the overall dataset is currently more correlative than can be deemed causative, it would be important to provide this data. Additionally, RNA PolIII Chip-Seq to corroborate reduced transcription rates/occupancy across the vector genome with different capsids would be useful. While these experiments may not be necessary across the manuscript, at least early/critical data would be significantly strengthened.

3. It is unclear why AAV-DJ was chosen as a control? This capsid does not appear to bear homology to LK-03 or variants thereof. While the capsid may have been selected as a positive control for liver (regardless of species) from earlier studies by this group, it seems that AAV3 (high sequence similarity to LK03) or AAV8 (which has been shown by this group and others to transduce murine liver preferentially over human hepatocytes in humanized mice) would serve as important and more relevant controls.

4. In Figure 4, Day 3 vs Day 15 differences in H3K4me in mouse liver are confusing. Why does LK03 show higher number of reads for activation marks at later time intervals? In particular, Figure 4b-d does not appear to be consistent with the observations from earlier. Figure 4b shows higher transduction by LK03

in mouse liver (day 15/day 3 ratio); Figure 4c is based on RT-qPCR which shows the opposite trend (unlike Figure 3, which seems to show qPCR data); Figure 4d shows LK03 with all read numbers higher than the rest of the samples and it is unclear why this data is shown/used for normalization. Figure 4e CPM is inconsistent and much lower (0-0.1) compared to other figures, where CPM appears to be between 1-5 and with in vivo appears to be 0-1(Extended Fig 4).

5. Lack of differences in promoters are shown in extended figure 1b, but both examples are ubiquitous promoters (EF1/CMV). Due to the liver focus of the study, examples of liver-specific promoters should be assessed to strengthen the overall findings.

Reviewer #2 (Remarks to the Author):

In this paper, Adriana Gonzalez-Sandoval, Mark Kaye, and colleagues present the finding that the reduced transduction of mouse cells in vitro and mouse liver in vivo by AAV-LK03 is due to differences in epigenetic marks on the vector genome compared to two other variants (AAV-DJ and the novel AAV-AM).

I am very excited to see this study, which chips away at the enigmatic and understudied question of how the AAV capsid affects transcription. This represents an expansion of our understanding of the basic biology of AAV, an area of critical importance. It also introduces the idea that differences in chromatin remodeling even between very similar variants can account for species-specific differences in transduction. The contribution of the AAV-AM capsid, which differs from AAV-LK03 by only a single amino acid insertion but has dramatically increased transcription in mouse cells, is very interesting. This paper does not propose a mechanistic basis for how the capsid influences chromatin remodeling, so the overall significance is reduced and this puzzle remains partially unsolved. I have reservations about the statistical tests and reporting as well as the data visualization in this paper that should be addressed before it is accepted for publication. Overall I believe that this study represents a good contribution to the field and the scientific approach is thorough.

Major comments:

1. I am not sure that the way the significance of this paper is framed is most effective or accurate. The title of the paper implies that capsid-mediated changes to the chromatin state broadly controls host range, but the paper focuses on only one variant and does not propose a mechanism. While it's not true that this is always the driving force behind species-specific differences in transduction efficiency between variants (PHP.B/eB are a clear example of the role of receptor binding), it seems to be the case at least for AAV-LK03. However, from my perspective the real finding of this study is that the AAV capsid influences transcription after second-strand synthesis through the chromatin remodeling pathway,

which is quite a deep and beautiful result! While you were unfortunately slightly beaten to the punch by Das et al.'s JVI paper earlier this year, your study is a much more thorough investigation of the role of the capsid specifically while theirs focuses more on the general case of epigenetic regulation of rAAV genomes. I encourage you to emphasize more how this study contributes to our understanding of basic AAV biology rather than just the implications for species-specific differences in transduction.

2. The information provided in the paper on the statistical tests is unclear and insufficient, and I am not sure that the correct tests were used in all cases.

a. In each of the figure legends, please specify which test is being used and what n is.

b. In both the methods section and the figure legends, please describe what kind of correction for multiple comparisons is being applied. Most of your quantitative data requires a correction of some sort unless you are only performing a single t test.

c. There are many instances (all of main figure 1, 3B, 3C, 3F, 3G, all of extended data figure 1; extended data 3E, 5A, 5B) where data is presented and it's not clear whether a statistical test just wasn't done or if it was done but was not significant. If statistical tests were not performed, please do so. If they were performed but not significant, there are some places where it would be clearer to insert a line with "ns" above it to distinguish it from the significant results.

d. The methods section only mentions t tests, which are valid for some of the datasets in the paper but not all of them. Instances where more than two groups are being compared, such as in figure 2, should be tested with ANOVA.

3. Can you account for the high variability in the data in figure 3C? I think this experiment should be repeated with a greater number of replicates.

4. In both the abstract and discussion you propose that the uncoating process is tied to capsid-dependent changes in epigenetic markers, but it is not clear where this is supported in the paper. Please explain further.

5. There are several places in the paper where the writing is vague or unclear and obscures the science. For example, starting line 234: "Capsid proteins being important for transduction efficiency have been slowly gaining some spotlight in other contexts as well, where various capsids have differential effects on expression from various promoters used in the central nervous system". Without additional context or explanation, it's not clear what this statement adds to the discussion. Other grammatical errors, such as "AAV genomes can become chromatinization" (line 249) and comma errors throughout the paper should be addressed. Careful attention to the clarity and specificity of the writing will allow the science to shine.

6. A more thorough set of references along with explicit textual connections between this study and key references will strengthen the paper and better contextualize it in the field. For example, the 2014 Salganik paper¹ (reference 18) is a critical piece of background for this study as it was the first to report that the capsid can play a role in transcription after the second-strand synthesis stage, but this paper is only mentioned in passing.

7. I recommend reading and incorporating the 2015 PLoS Pathogens paper by Schreiber et al² (I am not affiliated with this paper or the authors, just a fan). They found that two components of the U2 snRNP

spliceosome, PHF5A and SF3B1, interact with a broad range of AAV capsid serotypes, and PHF5A additionally interacts with the AAV genome itself. These two factors inhibit AAV transcription after the second-strand synthesis step in a manner independent of canonical U2 snRNP spliceosome function. Both of these factors have been proposed to moonlight in the chromatin remodeling pathway: PHF5A has a histone reader domain and alters the deposition of histone modifications in breast cancer,³ and SF3B1 interacts with histone H3 tails⁴ and the WSTF-SNF2h chromatin remodeling complex.⁵ While investigating the role of these factors is probably outside the scope of this study, connecting these dots and finding the mechanistic basis for the role of the capsid in transcription would monumentally expand our collective understanding of AAV biology. It would be great to get into this with a bit more specificity in the discussion, the Schreiber paper is just a starting point and suggestion.

Minor comments:

1. Please make sure you are clear that the species-specific difference in transduction by AAV-LK03 that you are referring to is in the liver. This could be a bit more clear throughout the paper.
2. In the data processing subsection of the methods section, it should read “replicates” not “replicas”.
3. Please convert bar graphs to scatter plots with mean and error bars or box and whisker plots and increase the size/visibility of individual data points and significance asterisks.
4. Please convert the y axis to linear scale so the data is more visible for the following figure panels: 1C, 1D, 1E, 1H, 1I, 3G, extended data 1C.
5. The range of the y axis on several plots (4B, 4C, 4D, extended data 5C) is unnecessarily large; please make the graphs so that the data takes up most of the range of the y axis for best interpretability.
6. In figure 3C, it appears that the error bars are not centered around the mean, which should always be the case for the standard error.
7. Please describe what software or tool was used to produce figure 3A.
8. There are several places in the text where you discuss data that is not marked as statistically significant in the figure and use language like “a similar proportion” (line 108). Please use more explicit and quantitative language to describe data in cases where statistical tests were applied.
9. The data in figure 4 seems to be based on a sample size of 1. While I don’t think that this information is the key finding of the paper, please be extremely explicit about the limitations of this experiment and that you are not reporting statistically significant results in the text.
10. Extended data figure 3A is too small to be legible.

References:

1. Salganik, M. et al. Adeno-Associated Virus Capsid Proteins May Play a Role in Transcription and Second-Strand Synthesis of Recombinant Genomes. *Journal of Virology* 88, 1071–1079 (2014).
2. Schreiber, C. A. et al. An siRNA Screen Identifies the U2 snRNP Spliceosome as a Host Restriction Factor for Recombinant Adeno-associated Viruses. *PLOS Pathogens* 11, e1005082 (2015).
3. Zheng, Y.-Z. et al. PHF5A Epigenetically Inhibits Apoptosis to Promote Breast Cancer Progression. *Cancer Research* canres.3514.2017 (2018) doi:10.1158/0008-5472.CAN-17-3514.
4. Heo, K. et al. Isolation and Characterization of Proteins Associated with Histone H3 Tails in Vivo. *J. Biol. Chem.* 282, 15476–15483 (2007).
5. Cavellán, E., Asp, P., Percipalle, P. & Farrants, A.-K. Ö. The WSTF-SNF2h Chromatin Remodeling Complex Interacts with Several Nuclear Proteins in Transcription. *J. Biol. Chem.* 281, 16264–16271 (2006).

REVIEWER COMMENTS

We were happy to learn there was general enthusiasm for this manuscript. We thank the reviewers for their helpful suggestions and as a result we performed quite a few additional experiments and re-wrote portions of the manuscript most notably the discussion.

A brief summary of the major added data.

- 1) Luciferase expression and luciferase mRNA quantification (Extended Data Fig. 3b and 3c) for AAV-AM, AAV-LK03, and AAV-LK03 265insertionT transduction experiments.
- 2) New/repeat experiments in Extended Data Fig. 1 (e-g) quantifying AAV-LK03 vs AAV-DJ luciferase activity, mRNA expression and genome copy number in human and mouse cells.
- 3) New/repeat experiments comparing different promoter and transgenes for transduction quantification with AAV-DJ and LK03 transduction in mouse and human cells (Extended Data Figure 1 h).
- 4) Luciferase mRNA expression data for AAV-AM and LK03 transduction in human and mouse cells (Fig. 3b).
- 5) Tn5 normalization included for all the in vivo studies (Fig. 3 i, Fig. 4 d, Extended Data Fig. 6 a-c, Extended Data Fig. 8 c, f, e)
- 6) Cut n Tag study using the anti-RNA pol II antibody (Extended Data Fig. 7).
- 7) Extensive statistical evaluation for each study.

The specific comments to each query are summarized below.

Reviewer #1 (Remarks to the Author):

This manuscript by Gonzalez-Sandoval and colleagues is focused on evaluating the interplay between AAV capsids and their genomes, which in turn is hypothesized to impact transduction efficiency. Briefly, the authors provide evidence that a capsid (AAV-LK03) isolated from primate liver (humanized mouse model) tissue through directed evolution, shows decreased transduction in murine hepatocytes and imparts distinct chromatin marks on the vector genome. Depleted histone marks generally associated with transcription are observed in case of the original capsid only in mice, but a single point mutation is associated with active marks on the vector genome presumably restoring transduction in both species. Overall, the study presents novel observations/phenomena potentially linking the capsid and epigenetic modifications of the vector genome. The approach utilized is balanced, toggling skillfully between capsid modifications and extensive interrogation of molecular fate of vector genomes in human and mouse liver models. The manuscript captures the essence of these seemingly unrelated events in AAV biology and their novelty as well as briefly outlines implications for human gene therapy. However, significant gaps in the data, experimental design/controls and overall conclusions are noted as outlined below.

1. First, the authors confirm that LK03 shows lower transgene and mRNA transcript expression in human hepatocarcinoma cells in vitro over the mouse cell line. This appears to correlate with lower H3K4me and H3K27Ac signal in murine vs human cell lines, and no change is seen with H3K9me3 and H3K27me. However, the current normalization method seems to be focused on read depth using a Tn5

normalizer method. No controls using isotype matched IgG have been shown/evaluated. Normalization to the latter signal will be important in addressing the extent of non-specific reads throughout the manuscript. This is particularly important since ITR related reads have been removed from the overall calculations (due to overrepresentation).

We have done the IgG controls and while these showed low background reads for the genomic DNA, as it is expected, the IgG- control AAV reads gave high background and it was variable between the species. We believe this may be due to something in the IgG control polyclonal antibodies that may be binding to the AAV in a non-specific manner. Because we are using different AAVs and cells from different species, we elected not pursue this type of control. In addition, while IgG controls have been used for genomic host DNA analyses, they have been used less often over the last few years. This was confirmed with discussions with Howard Chang and Will Greenleaf at Stanford. Will Greenleaf's group suggested the Tn5 normalization controls. Since the original submission, we have completed the Tn5 normalization for all the CnT experiments. Such controls reduce non-specific reads in the input cells used in each individual experiment.

2. Second, the authors note that a single G insertion in LK03 (AAV-AM) increases transduction concomitant with H3K4me and H3K27Ac reads increasing both in vitro and in vivo. However, no mRNA levels are shown in vitro or in vivo (Figure 3). Since the overall dataset is currently more correlative than can be deemed causative, it would be important to provide this data.

These data were added to the revised manuscript and are now included in Figure 3 (see Fig. 3b, g, Extended Data Fig. 3c)

Additionally, RNA PolIII Chip-Seq to corroborate reduced transcription rates/occupancy across the vector genome with different capsids would be useful. While these experiments may not be necessary across the manuscript, at least early/critical data would be significantly strengthened.

These experiments were included in the revised manuscript in Extended Figure 7. These data corroborate the expression data as the occupancy parallels, the mRNA and protein levels.

3. It is unclear why AAV-DJ was chosen as a control? This capsid does not appear to bear homology to LK-03 or variants thereof. While the capsid may have been selected as a positive control for liver (regardless of species) from earlier studies by this group, it seems that AAV3 (high sequence similarity to LK03) or AAV8 (which has been shown by this group and others to transduce murine liver preferentially over human hepatocytes in humanized mice) would serve as important and more relevant controls.

AAV-DJ was selected as the control because we wanted one capsid that was known to provide relatively equal levels of transduction (measured by transgene expression) in the test cells and mouse liver in vivo. AAV-DJ fits that requirement. AAV3

behaves similarly to LK03 and would likely be redundant and give similar results to AAV-LK03. AAV8 transduces most cells in culture very poorly requiring extremely high MOIs to get any signal in most culture cells including the ones used in these studies and we believe this would not be an appropriate control.

4. In Figure 4, Day 3 vs Day 15 differences in H3K4me in mouse liver are confusing. Why does LK03 show higher number of reads for activation marks at later time intervals? In particular, Figure 4b-d does not appear to be consistent with the observations from earlier. Figure 4b shows higher transduction by LK03 in mouse liver (day 15/day 3 ratio); Figure 4c is based on RT-qPCR which shows the opposite trend (unlike Figure 3, which seems to show qPCR data); Figure 4d shows LK03 with all read numbers higher than the rest of the samples and it is unclear why this data is shown/used for normalization. Figure 4e CPM is inconsistent and much lower (0-0.1) compared to other figures, where CPM appears to be between 1-5 and with in vivo appears to be 0-1 (Extended Fig 4).

We revised the figures after performing the Tn5 normalization as we had previously only done for the in vitro studies. We hope the data are clearer that way now. Over time between 3 and 15 days, transgene expression goes up while the vector nuclear genome copy number either is similar or falls. The AAV-LK03 delivered vector copies fell the greatest amount (~100x). This is summarized in Fig. 4b-d. However, the copies that do remain are highly enriched for histones. Because the enrichment scores are corrected for genome copies, the remaining genomes are indeed highly enriched for histones even though transgene expression throughout the liver is low. We suggest the episomes that are stable and thus enable expression must form some sort of favorable structure after interacting with the host nuclear proteins (e.g. histones) and this process is much less efficient or slower in AAV-LK03 treated murine cells compared to AAV-DJ or AAV-AM. Thus, for AAV-LK03 delivered genomes many episomes get degraded resulting in low expression levels.

5. Lack of differences in promoters are shown in extended figure 1b, but both examples are ubiquitous promoters (EF1/CMV). Due to the liver focus of the study, examples of liver-specific promoters should be assessed to strengthen the overall findings.

We decided not to use a liver specific promoter because the expression from these promoters is rather weak in cultured hepatocytes and hepatoma cells—both murine and human derived. The species selectivity of AAV-LK03 holds true in non-hepatic cells as well (Lisowski et al., 2014 Nature). We added additional experiments using the EF1alpha and CMV promoter which are now provided in Extended Data Fig. 1h in the revised manuscript.

Reviewer #2 (Remarks to the Author):

In this paper, Adriana Gonzalez-Sandoval, Mark Kay, and colleagues present the finding that the reduced transduction of mouse cells in vitro and mouse liver in vivo by AAV-LK03 is due to differences in epigenetic marks on the vector genome compared to two other variants (AAV-DJ and the novel AAV-AM).

I am very excited to see this study, which chips away at the enigmatic and understudied question of how the AAV capsid affects transcription. This represents an expansion of our understanding of the basic biology of AAV, an area of critical importance. It also introduces the idea that differences in chromatin remodeling even between very similar variants can account for species-specific differences in transduction. The contribution of the AAV-AM capsid, which differs from AAV-LK03 by only a single amino acid insertion but has dramatically increased transcription in mouse cells, is very interesting. This paper does not propose a mechanistic basis for how the capsid influences chromatin remodeling, so the overall significance is reduced and this puzzle remains partially unsolved. I have reservations about the statistical tests and reporting as well as the data visualization in this paper that should be addressed before it is accepted for publication. Overall I believe that this study represents a good contribution to the field and the scientific approach is thorough.

Major comments:

1. I am not sure that the way the significance of this paper is framed is most effective or accurate. The title of the paper implies that capsid-mediated changes to the chromatin state broadly controls host range, but the paper focuses on only one variant and does not propose a mechanism.

We did not mean to imply that differences in the chromatin state broadly controls host range transduction (measured by transgene expression). To clarify this: (1) We changed the title; (2) Show a model in **Fig 4.**; (3) Changed the contents of the discussion based on the comments made by this reviewer (see below).

While it's not true that this is always the driving force behind species-specific differences in transduction efficiency between variants (PHP.B/eB are a clear example of the role of receptor binding), it seems to be the case at least for AAV-LK03. However, from my perspective the real finding of this study is that the AAV capsid influences transcription after second-strand synthesis through the chromatin remodeling pathway, which is quite a deep and beautiful result! While you were unfortunately slightly beaten to the punch by Das et al.'s JVI paper earlier this year, your study is a much more thorough investigation of the role of the capsid specifically while theirs focuses more on the general case of epigenetic regulation of rAAV genomes. I encourage you to emphasize more how this study contributes to our understanding of basic AAV biology rather than just the implications for species-specific differences in transduction.

We are grateful for those helpful suggestions and included those publications in the discussion of the revised discussion. A discussion around the implications for AAV biology especially based on this reviewer's comments can be found in the revised manuscript.

2. The information provided in the paper on the statistical tests is unclear and insufficient, and I am not sure that the correct tests were used in all cases.

a. In each of the figure legends, please specify which test is being used and what n is.

b. In both the methods section and the figure legends, please describe what kind of correction

for multiple comparisons is being applied. Most of your quantitative data requires a correction of some sort unless you are only performing a single t test.

c. There are many instances (all of main figure 1, 3B, 3C, 3F, 3G, all of extended data figure 1; extended data 3E, 5A, 5B) where data is presented and it's not clear whether a statistical test just wasn't done or if it was done but was not significant. If statistical tests were not performed, please do so. If they were performed but not significant, there are some places where it would be clearer to insert a line with "ns" above it to distinguish it from the significant results.

d. The methods section only mentions t tests, which are valid for some of the datasets in the paper but not all of them. Instances where more than two groups are being compared, such as in figure 2, should be tested with ANOVA.

This is a very important point, we wish to apologize for this lack of due diligence on our end. There was some ambiguity in which tests were done in the original manuscript as most of the description of the statistics was in the methods. We have extensively revised our statistical analyses and clearly stated the statistical test used and p value in the figure legend. Moreover, we extended the statistics for some of the studies not analyzed in the original manuscript. Overall, the statistical descriptions are now complete and more clearly denoted in the figures.

3. Can you account for the high variability in the data in figure 3C? I think this experiment should be repeated with a greater number of replicates.

There is some variability in Fig 3C. However, the most important comparison in this figure was nuclear genome copies in AAV-LK03 delivery in mouse and human cells. The AAV-Lk03 comparison was done in two separate sets of experiments: one with AAV-DJ (Fig, 1 a-d) and one with AAV-AM (Fig. 3 a-c) . A similar set-up was performed for the in vivo studies due to the sequence of events as the project was evolving (day 3 data are in Fig. 1 f-l for AAV-LK03 and AAV-DJ, Fig. 3 f-h for AAV-LK03 and AAV-AM, day 15 data are in Extended Data Fig. 8 a-b). All those data sets contain nuclear copy number quantification data and are inherently variable due to the expansive sample manipulation (see Methods section for the protocol). While variability for those data sets is high the trend is consistent.

4. In both the abstract and discussion you propose that the uncoating process is tied to capsid-dependent changes in epigenetic markers, but it is not clear where this is supported in the paper. Please explain further.

The fact that the same genomes when delivered by different capsids into murine cells result in vastly different transduction efficiency despite similar levels of cell entry implies that the process responsible for the observed differences occurs at some time during uncoating—it may occur early or late in the process, but by definition different capsids result in different epigenetic marks and it has to occur at some during or just after the genomes are released. The only variable is the capsid, so it somehow has to be associated with release of the genome. Moreover, we are careful to use terms like "this suggests ..." rather than making absolute statements.

5. There are several places in the paper where the writing is vague or unclear and obscures the science. For example, starting line 234: "Capsid proteins being important for transduction efficiency have been slowly gaining some spotlight in other contexts as well, where various capsids have differential effects on expression from various promoters used in the central

nervous system". Without additional context or explanation, it's not clear what this statement adds to the discussion. Other grammatical errors, such as "AAV genomes can become chromatinization" (line 249) and comma errors throughout the paper should be addressed. Careful attention to the clarity and specificity of the writing will allow the science to shine.

We have rephrased these statements.

6. A more thorough set of references along with explicit textual connections between this study and key references will strengthen the paper and better contextualize it in the field. For example, the 2014 Salganik paper¹ (reference 18) is a critical piece of background for this study as it was the first to report that the capsid can play a role in transcription after the second-strand synthesis stage, but this paper is only mentioned in passing.

See response below critique 7.

7. I recommend reading and incorporating the 2015 PLoS Pathogens paper by Schreiber et al² (I am not affiliated with this paper or the authors, just a fan). They found that two components of the U2 snRNP spliceosome, PHF5A and SF3B1, interact with a broad range of AAV capsid serotypes, and PHF5A additionally interacts with the AAV genome itself. These two factors inhibit AAV transcription after the second-strand synthesis step in a manner independent of canonical U2 snRNP spliceosome function. Both of these factors have been proposed to moonlight in the chromatin remodeling pathway: PHF5A has a histone reader domain and alters the deposition of histone modifications in breast cancer,³ and SF3B1 interacts with histone H3 tails⁴ and the WSTF-SNF2h chromatin remodeling complex.⁵ While investigating the role of these factors is probably outside the scope of this study, connecting these dots and finding the mechanistic basis for the role of the capsid in transcription would monumentally expand our collective understanding of AAV biology. It would be great to get into this with a bit more specificity in the discussion, the Schreiber paper is just a starting point and suggestion.

We are aware of these studies but also have some constraints on the length of the discussion. Nonetheless, we believe it was a good idea to expand on this and rewrote the Discussion to include an expanded section as suggested in critiques 6 and 7.

Minor comments:

1. Please make sure you are clear that the species-specific difference in transduction by AAV-LK03 that you are referring to is in the liver. This could be a bit more clear throughout the paper. Done

2. In the data processing subsection of the methods section, it should read "replicates" not "replicas". Done

3. Please convert bar graphs to scatter plots with mean and error bars or box and whisker plots and increase the size/visibility of individual data points and significance asterisks. We tried several ways to display the data and still prefer bar graphs. However, we worked on a better visual appearance of our figures.

4. Please convert the y axis to linear scale so the data is more visible for the following figure panels: 1C, 1D, 1E, 1H, 1i, 3G, extended data 1C. We changed the graphs accordingly.

5. The range of the y axis on several plots (4B, 4C, 4D, extended data 5C) is unnecessarily large; please make the graphs so that the data takes up most of the range of the y axis for best interpretability. We updated all graphs in Fig. 4d and Extended Data Fig. 8c due to the display of Tn5 normalized in vivo data in the revised version of the manuscript. Furthermore we changed the y axis for the Luciferase activity and nuclear copy number ratios. We also

discovered a mistake in Fig. 4a which we corrected.

6. In figure 3C, it appears that the error bars are not centered around the mean, which should always be the case for the standard error. We display the data as mean (column) with SEM (standard error of the mean) error bars.

7. Please describe what software or tool was used to produce figure 3A. This model was now moved to Extended Data Fig. 3e and the software used is included in the Methods section.

8. There are several places in the text where you discuss data that is not marked as statistically significant in the figure and use language like “a similar proportion” (line 108).

Done

Please use more explicit and quantitative language to describe data in cases where statistical tests were applied. Done

9. The data in figure 4 seems to be based on a sample size of 1. While I don't think that this information is the key finding of the paper, please be extremely explicit about the limitations of this experiment and that you are not reporting statistically significant results in the text. Fig. 4b-d show the ratios of the average values from experiments done multiple times as described. The original data displaying each replicate can be found in Fig. 1f, 1i, Fig. 3f, 3h, 3i, Extended Data Fig. 6a, Extended Data Fig. 8a, 8b, 8e. We have re-written and clarified these analyses in the text.

10. Extended data figure 3A is too small to be legible. We re-sized this figure and placed it in Extended Data Fig. 3a.

REVIEWERS' COMMENTS

Reviewer #1 (Remarks to the Author):

The authors have addressed any concerns and questions raised. The manuscript is a significant advancement in the field.

Aravind Asokan

Reviewer #2 (Remarks to the Author):

It was a pleasure to read the authors' revised manuscript. While this was already a thorough and skillfully done study, the revisions make the manuscript substantially more clear and place it in much better context within the field. The additions to the discussion section are excellent and hopefully will generate broader interest in the study and in the role of capsid-dependent epigenetic marks on AAV episomes in species-specific transduction patterns and AAV transduction in general.

In the initial submission I had many concerns about the statistical methods and reporting in the paper. I am happy that these concerns have been addressed and the figures themselves as well as their descriptions in the legends are much clearer. I am not sure of the specific requirements for this journal; they may ask you to include a description of the statistical analyses performed in the methods section, but what you have included in the figure legends is clear and concise. These changes have substantially increased my confidence in the authors' results and analyses.

Unlike the other reviewer, I do not see an issue with using AAV-DJ as a control specifically given the inclusion of AAV-AM (a striking result!) or using only ubiquitous promoters. Additional promoters would have been nice but I don't think they are critical. The addition of extended data figure 7 at the other reviewer's request is very nice and I agree that it strengthens the manuscript's findings, as do the addition of mRNA level data in figure 3.

Overall I am pleased with the extra work and revisions that the authors have performed and feel that this study is ready for publication.